# Garnet microstructures suggest ultra-fast decompression of ultrahigh-pressure rocks

Cindy Luisier ®[1] ✉, Lucie Tajčmanová ®[2], Philippe Yamato ®[1,3] & Thibault Duretz[1]

Plate tectonics is a key driver of many natural phenomena occurring on Earth, such as mountain building, climate evolution and natural disasters. How plate tectonics has evolved through time is still one of the fundamental questions in Earth sciences. Natural microstructures observed in exhumed ultrahigh-pressure rocks formed during continental collision provide crucial insights into tectonic processes in the Earth's interior. Here, we show that radial cracks around $SiO_2$ inclusions in ultrahigh-pressure garnets are caused by ultrafast decompression. Decompression rates of at least 8 GPa/Myr are inferred independently of current petrochronological estimates by using thermo-mechanical numerical modeling. Our results question the traditional interpretation of fast and significant vertical displacement of ultrahigh-pressure tectonic units during exhumation. Instead, we propose that such substantial decompression rates are related to abrupt changes in the stress state of the lithosphere independently of the spatial displacement.

The understanding of plate tectonics processes, such as mountain building, usually relies on petrochronological data inferred from microscale observations in exhumed rock samples[1–3]. The microscale observations reveal relics of mineral transformations related to changes in pressure ($P$) and temperature ($T$) during the tectonic evolution experienced by the rock samples. In that sense, ultrahigh-pressure (UHP) rocks—i.e., rocks that underwent extreme $P$ conditions in the deep Earth, documented by the formation of high-pressure silica ($SiO_2$) or carbon polymorphs (coesite or diamond, respectively)—are the key witnesses of deep processes during burial and exhumation of the Earth's lithosphere. However, the mechanisms of exhumation of UHP units remain one of the most challenging yet unsolved questions that keep on feeding controversies over the decades[4–8]. One of the first evidence of UHP metamorphism on Earth was provided by Chopin[9], based on the observation of coesite relics in pyrope garnets from the Dora-Maira whiteschists (Brossasco-Isasca unit, Western European Alps). Thanks to its emblematic status, the UHP Dora Maira whiteschist has been the most studied UHP rock worldwide and can serve as a proxy for refining our knowledge of the burial and exhumation cycle[4,9–22] (Fig. 1a). The $P$-$T$ and time ($t$) petrochronological data provide robust constrains for building up geodynamic scenarios. The $P$-$T$-$t$ dataset (Fig. 1a) indicates that the coesite-bearing Dora-Maira

whiteschists reached peak $P$-$T$ conditions of 4.3 GPa and 730 °C[16] at around 35.1 ± 0.9 Ma[4] and then a first retrograde stage of 1.1 GPa and 600 °C[16] at 32.9 ± 0.9 Ma[4]. Traditionally, the geochronological and petrological data were also used to estimate an average exhumation rate that corresponds to the velocity at which the rocks are exhumed toward the surface. The time is then given by geochronological data, and the variation of depth is computed assuming that pressure is directly proportional to depth (i.e., $P$ is considered lithostatic). Hence, for Dora Maira, an average exhumation rate of ~5 cm/year can be estimated[4]. Several exhumation mechanisms of UHP rocks have been proposed in the past based on available petrological, structural and geochronological data[23–28]. However, exhumation from mantle depth involves displacement larger than 100 km, for which geological evidence remains scarce[7,29]. Alternatively, it has been suggested that the major part of the decompression of (U)HP rocks does not translate into a vertical displacement and rather corresponds to a rapid switch in the tectonic stress state within collision zones[30].

The iconic petrographic observations in the UHP rocks are the partial preservation of UHP coesite inclusions in garnet that were partially to completely turned into α-quartz - the low-pressure $SiO_2$ polymorph—during retrogression and the associated radial fractures in the garnet host. The relationship between the coesite to α-quartz phase

[1]Institute of Geosciences, Goethe University Frankfurt, Frankfurt am Main, Germany. [2]Institute of Earth Sciences, Heidelberg University, Heidelberg, Germany. [3]Univ Rennes, CNRS, Géosciences Rennes, Rennes, France. ✉ e-mail: cindy.luisier@gmail.com

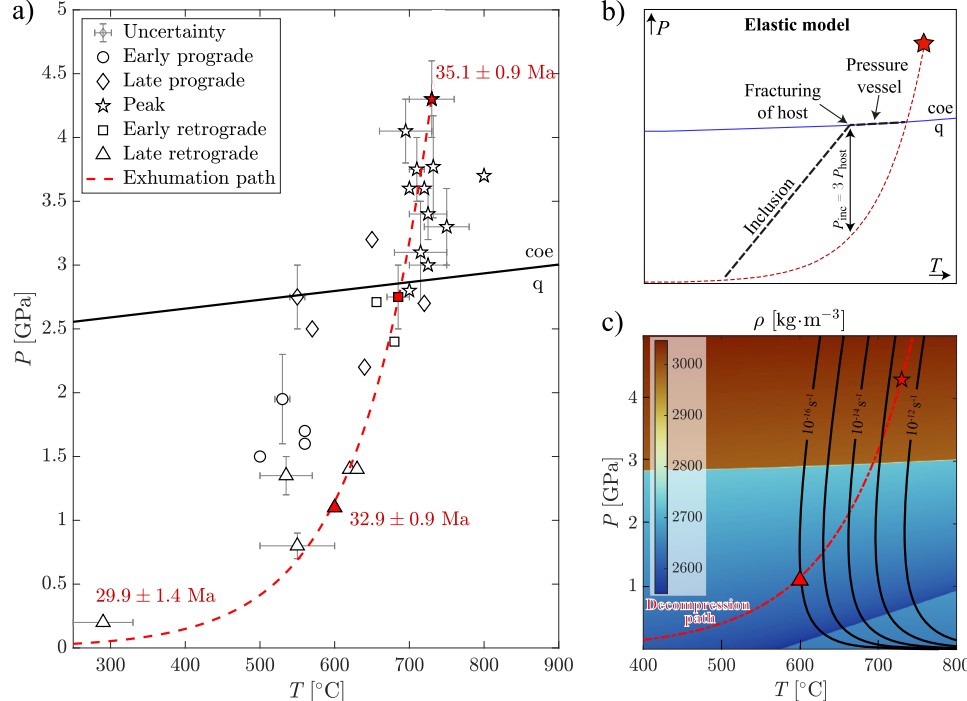

**Fig. 1 | Compilation of available petrochronological and mechanical data.**
**a** Summary of the pressure and temperature (*P-T*) data for the Dora Maira Massif. Age data for the peak metamorphic event (4.3 GPa, 730 °C)[4] and retrograde events[4,14] are also reported. The black line corresponds to the quartz (q)-coesite (coe) transition, and the red dashed line shows the retrograde path calculated based on the *P-T* data[16] (red symbols). Error bars indicate the minimum and maximum values of *P* and *T* estimates. **b** Sketch of the elastic model explaining the two stages of pressure evolution of an inclusion (black dashed line) entrapped in a garnet during the retrograde *P-T* path (red dashed line). The first stage (along the coesite-quartz transition) corresponds to the pressure vessel effect. The second stage of pressure decrease occurs after garnet fracturing, when the threshold of $P_{inc} = 3 \times P_{host}$ is reached (see text for explanation). **c** Colormap of $SiO_2$ density in $kg \cdot m^{-3}$. The red dashed line and symbols are as in (**a**). Black curves show the brittle-ductile transition for different deviatoric strain rates (brittle and ductile on the left and right side of the curve, respectively) computed using the garnet flow law[50].

transition and the development of radial fractures in garnet may provide precious constraints on the exhumation of UHP units and the associated geodynamic evolution[31]. Previous studies focusing on the preservation of coesite have traditionally explained it by the pressure vessel effect, which relies on the differential volumetric expansion of the inclusion and host minerals during decompression[31,32] (Fig. 1b). Garnet is a strong mineral that cannot accommodate the volume increase induced by the coesite to α-quartz transformation. Therefore, high pressure is maintained on the expanding inclusion, and the reaction is prevented from running to completion. However, these models only consider the elastic behavior of garnet (i.e., no permanent deformation and hence no fracturing) and therefore, radial fracturing was traditionally constrained to occur as soon as the pressure difference between the inclusions ($P_{inc}$) and the garnet host ($P_{host}$) overcomes a factor 3, i.e., $P_{inc} \geq 3P_{host}$ (Fig. 1b). Alternatively, the growth of pre-defined cracks has been modeled by combining elasticity and fracture mechanics[35]. Previous models usually used simple geometries involving either a single spherical inclusion in a spherical host[34] or faceted inclusion[35] in a host. The role of viscous relaxation and plastic deformation has been shown to affect the pressure distribution in decompressing inclusion-hosts systems[36]. However, such analysis has never been quantitatively applied to investigate the fracturing of coesite-bearing UHP garnets.

Recent improvements in computing tools enable us to account for complex geometries (e.g., multiple mineral phases, digitalized thin sections). In contrast with past studies, elastic, viscous and plastic properties of the phases are now also incorporated, which allows for consistently simulating the coupling between reaction-induced volume changes and deformation[37]. Furthermore, recent studies have shown that UHP rocks, such as eclogites and, in particular, the mineral garnet, can deform in a brittle manner at high *P* conditions,

where they are expected to be viscous[37,38]. In fact, the brittle-ductile transition (BDT) of garnet for a whole range of strain rates intersects the decompression path of the UHP rocks such as the Dora Maira Massif (Fig. 1c). Interestingly, the brittle behavior of garnet can be used to constrain the range of decompression rates required to explain the petrographic observations. It is important to highlight that the term decompression path used here is purely related to the pressure decrease, i.e., independent of any spatial displacement.

Here, we combine mineral-scale thermo-mechanical numerical models with microstructural observations to quantify the geological conditions controlling radial fracture development in garnet at UHP metamorphic conditions in an inclusion-host-matrix system undergoing a large density change. Our results capture the typical patterns of radial microcracks in garnet surrounding retrogressed coesite inclusions. They challenge the traditional interpretation of a late-stage fracturing of the Dora Maria garnets based on the elastic model and show that, provided significant decompression rates, fracturing is induced by the large shear stress generated during volume change in the $SiO_2$ inclusions at the early stages of the phase transition.

## Results
### The Dora Maira microstructures
We selected a sample from the famous UHP Dora Maira whiteschists, consisting of garnet, phengite, kyanite, talc, quartz/coesite and rutile (Fig. 2a). The $SiO_2$ in the matrix is quartz and shows a polygonal texture, suggesting that it formed after retrogression from coesite[9], whereas the $SiO_2$ inclusions in garnet are either fully turned into quartz or consist of coesite rimmed by quartz. Two types of microcracks are observed in garnet: (1) a set of parallel straight fractures that have the same orientation in all garnets (Fig. 2a) and (2) sets of radial fractures

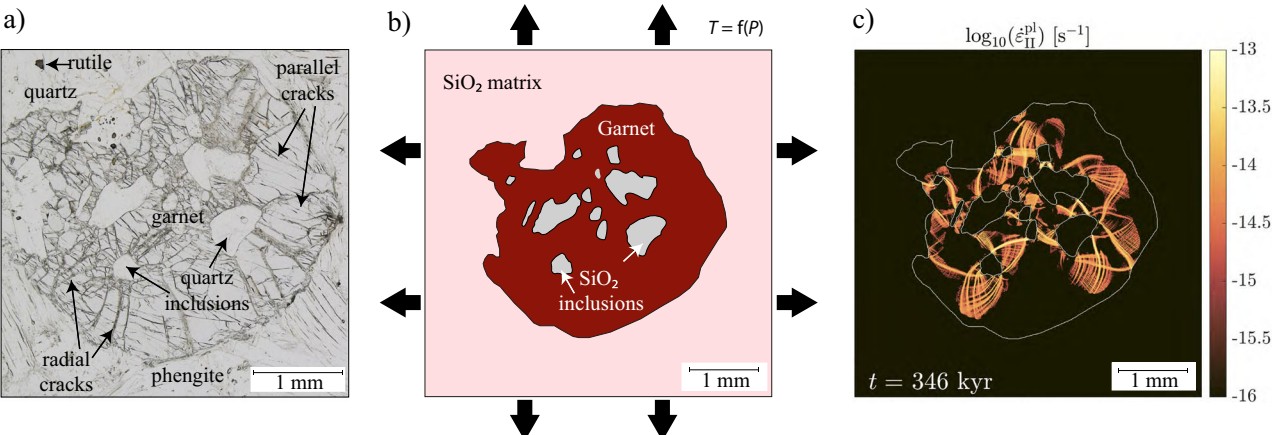

**Fig. 2 | Microscopic observations, model configuration and reference result.**
**a** Microphotograph of the sample in plane-polarized light showing one garnet with several SiO$_2$ (quartz) inclusions surrounded by radial cracks. The radial cracks are slightly bent. The matrix around garnet consists of phengite, quartz and rutile. **b** Model setup consisting of a garnet with several SiO$_2$ inclusions in a matrix of SiO$_2$. A constant dilation rate is applied to the box to simulate the decompression from the initial conditions of 4.3 GPa and 730 °C[16]. Pressure is computed, and temperature depends on the pressure following: $T = (1/(1.02 \times 10^{-2})) \cdot \ln(P/(2.523 \times 10^6)) + 273.15$. **c** Colormap showing the logarithm of the plastic strain rate (s$^{-1}$) after 346 kyr of decompression to 0.5 GPa and 500 °C (dilation rate: $10^{-14}$ s$^{-1}$, mean decompression rate: 11 GPa/Myr). Dilatant shear bands concentrate into radial and curved zones, shown in orange, mimicking the radial fractures around SiO$_2$ inclusions.

spreading from the SiO$_2$ inclusions (Fig. 2a). The latter form networks of fractures that either linkup to other SiO$_2$ inclusions or spread toward the sides of the garnet and eventually connect with the matrix. The fracture density is higher at the inclusion's corners as well as in the linkup regions than on the curved sides of inclusions, as documented and discussed in detail by Whitney et al. [35].

## Thermo-mechanical numerical modeling

A full description of the code used in this study is provided in the "Methods" section, together with a detailed presentation of the model setup (also summarized in Fig. 2b). The use of visco-elasto-plastic rheological models allows for the emergence of dilatant frictional-plastic shear bands in the garnet. These structures share similarities with radial microcracks observed in natural samples and are further qualified as radial fractures. Figure 2c shows the colormap of the plastic strain rate at the conditions where the matrix reaches 1 GPa (i.e., after 3.3 GPa of decompression). The high strain rate focuses on thin localized bands radiating from the SiO$_2$ inclusions and propagating between inclusions as well as toward the garnet rim. The patterns produced by the shear bands are in good agreement with the observed radial fractures in natural samples: (1) the shear bands are either straight or bent, and (2) the density of the shear bands is higher at the inclusion's corners, as expected based on previous studies[35].

The results of the thermo-mechanical numerical models are presented in Fig. 3. During the first stage of decompression, starting in the coesite stability field at 4.3 GPa and 730 °C, the matrix, garnet and inclusions undergo a progressive decrease in $P$ and $T$ until reaching the conditions of the phase transition (i.e., 2.95 GPa and ~700 °C; Fig. 3a). First, the matrix SiO$_2$ undergoes the coesite/α-quartz phase transition. At a constant dilation rate, the transition is not immediate and acts as a pressure buffer in the matrix, which remains at the pressure of the phase transition (i.e., 2.95 GPa, Fig. 4a, b). The SiO$_2$ inclusions remain in the coesite field as the volume increase in the matrix pressurizes the garnet and inclusions. This effect maintains the pressure of inclusions slightly above the level of the phase transition (Fig. 3b). When the entire matrix SiO$_2$ is turned into quartz, decompression of the garnet and inclusions resumes. At this stage, the decompression paths of the garnet, matrix and inclusions split. The decompression in the matrix progresses normally with a continuous $P$-$T$ decrease along the decompression path, and the coesite in the inclusions can start the phase transition (Fig. 3c). At this point, the geometry of the inclusions as well as the dilation rate control the stress and pressure evolution in

the garnet. The dilation rate is the first-order parameter that controls the decompression rate (Fig. 4c) and that most affects the deformation style and the pressure field in the model. The relative amount of matrix and garnet exerts a minor, second-order influence on the resulting decompression rate (see Supplementary Fig. 1). Hereafter, we present the results based on different dilation rates.

For dilation rates higher than and equal to $10^{-14}$ s$^{-1}$, we achieve a decompression rate of at least 8.0 GPa/Myr (Fig. 4). The garnet behaves in a brittle manner so that the stress increase in the garnet surrounding the transforming inclusions reaches the yield stress as soon as quartz starts forming, which triggers the nucleation of radial fractures (Figs. 2c, 3b). Therefore, the first fractures form early on the decompression path when the matrix reaches a $P$ of ca. 2.6 GPa, i.e., soon after the phase transition in the matrix is accomplished (Fig. 3b). The first fractures form linkups between neighboring SiO$_2$ inclusions by propagating the fracture tips toward the closest inclusions. Then, newly formed radial fractures propagate from the inclusion-garnet boundary toward the garnet sides (Fig. 3c).

If the phase transition of SiO$_2$ inclusions is disabled (Fig. 4a), i.e., coesite density follows a linear equation of state during decompression, the onset of plastic yielding occurs at lower $P$-$T$ conditions (ca. 1.6 GPa). This confirms the role of the SiO$_2$ phase transition in building up stresses in the host garnet. The phase transition fosters plastic yielding at high pressure, while differential expansion alone would trigger yielding only at later stages.

For a dilation rate of $10^{-15}$ s$^{-1}$, the decompression rate reaches 1.0 GPa/Myr. Diffuse plastic deformation occurs at a late stage (~1.5 GPa) once all SiO$_2$ turned to quartz. If the dilation rate is lower than $10^{-15}$ s$^{-1}$, garnet behaves in a ductile manner from the onset of the decompression path until ca. 2.0 GPa. The activation of viscous creep prevents the development of large differential stress around the SiO$_2$ inclusions so that the formation of quartz is enhanced without fracturing and goes to completion. Overall, localized plastic deformation is observed only for dilation rates greater or equal to $10^{-14}$ s$^{-1}$ and decompression rate of at least 8.0 GPa/Myr (Fig. 4b), and the onset of fracturing occurs consistently between 2.5 and 2.8 GPa.

The model results reproduce the first-order characteristics (i.e., shape, dimensions, location) of the radial fractures observed in natural samples. However, in our simulations, the timing of fracturing and the conditions in which it occurs differ from previous studies. In our models, plastic yielding occurs in response to a stress increase at the interface between the SiO$_2$ inclusions and the garnet host due to the

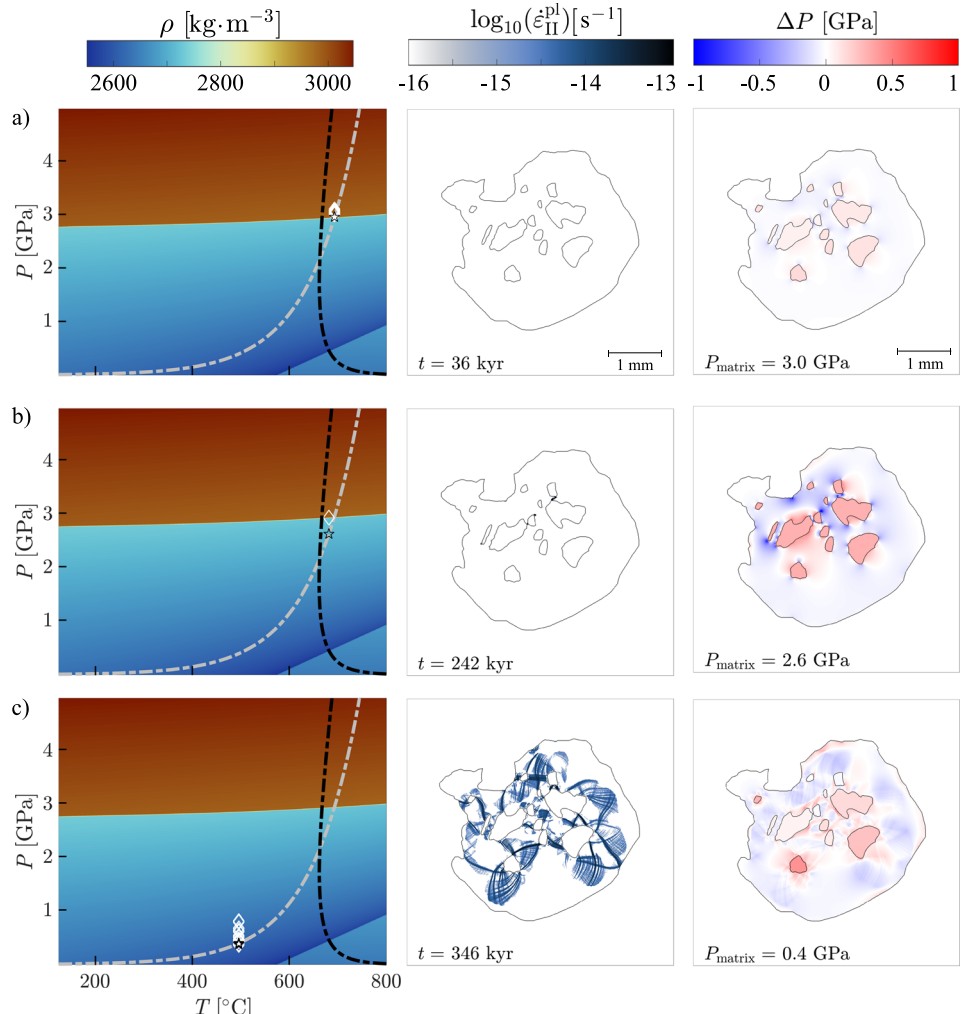

**Fig. 3 | Time evolution of the numerical model of inclusion-host decompression.** Model results for successive time steps for a decompression rate of 11 GPa/Ma. The first column shows *P-T* conditions of the matrix (black star) and each SiO$_2$ inclusion (white diamonds) along the retrograde path (gray dashed line). The black dashed line corresponds to the brittle-ductile transition for garnet at the strain rate of $10^{-14}$ s$^{-1}$, as in Fig. 1c. The second column shows the logarithm of the plastic strain rate. Blue areas correspond to the parts of the garnet that reached the plastic yield. The third column shows the relative pressure (in GPa) calculated as the pressure difference relative to the matrix. The time steps correspond to: **a** the SiO$_2$ inclusions, garnet and matrix SiO$_2$ reach the coesite-quartz transition; **b** the onset of garnet fracturing after the complete retrogression of the coesite in the matrix; **c** the matrix reaches a decompression stage of 0.4 GPa and 500 °C.

phase transition, whereas the elastic model assumes that fracturing is triggered by a pressure difference of a factor 3 between the inclusion and the garnet[33,34]. Moreover, in elastic models, fracturing occurs at lower temperatures (<400 °C) so that the kinetics of the coesite-quartz transformation is slow enough to prevent its total completion. In our model, the stress generated by the phase transition produces a pressure variation between the inclusions and garnet host that reaches ca. 2 GPa at the time when fracturing occurs. This happens much earlier and at higher *P-T* conditions (ca. 2.6 GPa, 680 °C in the matrix) than for elastic models. The complex geometry of inclusions also favors the development of shear stresses in the host in comparison with the purely spherical shapes considered in the previous elastic models. In our case, the preservation of coesite cannot be attributed to sluggish kinetics because the coesite-quartz transformation is almost instantaneous at high *T*[39,40], and another mechanism must be invoked to explain the preservation of coesite in the natural samples. To our knowledge, there is no available flow law for coesite. However, based on the observation of highly fractured coesite relicts in the SiO$_2$ inclusion cores, one could speculate that coesite is more resistant than quartz and hence behaves in a brittle manner, as garnet does, during the decompression. If this is true, the high mechanical strength of

coesite could allow it to remain at high pressure and prevent a total transformation into quartz. Furthermore, elastic models suggest that the pressure vessel effect is released soon after fracturing occurs. In our models, the pressure vessel effect remains after fracturing so that a pressure gradient of up to a factor 2 between the matrix and the inclusions persists toward the late stage of the decompression. The effect of a 3D geometry on the stress distribution around the inclusions remains to be investigated.

## Discussion
The decompression rates based on geochronological data of Dora Maira[4,41] between the peak conditions of 4.3 GPa and the retrograde conditions of 1.1 GPa translate into decompression rates varying between 0.8 GPa/Myr (maximum duration of ~4 Ma) and 8 GPa/Myr (minimum duration of ~0.4 Ma). Our results are thus consistent with the fastest decompression rates estimated from petrochronology and independently report the occurrence of ultra-fast decompression rates. Furthermore, the results suggest that radial fractures are formed early after the onset of decompression, i.e., after the phase transition has occurred in the matrix and while the phase transition is still occurring within the inclusions.

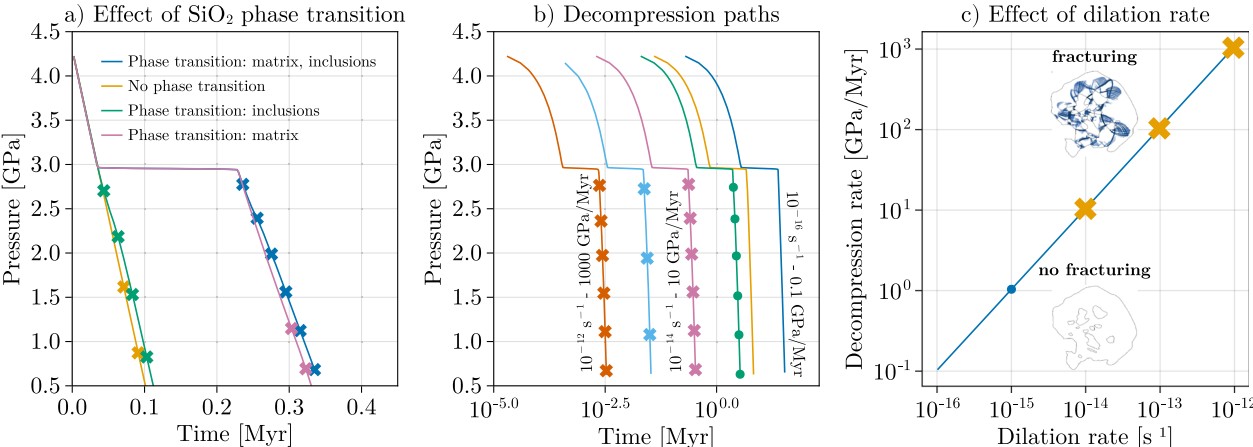

**Fig. 4 | Systematic investigation of host fracturing during decompression. a** Pressure (*P*) evolution of the matrix versus time (*t*) for a dilation rate of $10^{-14}$ s$^{-1}$. Simulations with phase transition activated either in the inclusions and matrix or only in the inclusions reach the yield stress at higher *P* conditions than simulations where phase transition is not activated or when phase transition occurs only in the matrix. The lines denote viscoelastic-dominated deformation, and crosses indicate the occurrence of localized frictional deformation. **b** Pressure evolution of the matrix versus time for different dilation rates. In contrast with runs with dilation rates higher than $10^{-14}$ s$^{-1}$ (red, light blue and purple lines with crosses), a dilation rate of $10^{-15}$ s$^{-1}$ induces diffuse plastic deformation (green line with circles), and dilation rates lower than $10^{-15}$ s$^{-1}$ caused no frictional plastic deformation (yellow, dark blue lines). **c** Relationship between dilation rate and the mean decompression rate, the blue dot indicates conditions for diffuse plastic deformation, and the yellow crosses indicate conditions for prominent shear banding.

Based on our results, a decrease in the decompression rate leads to the displacement of the BDT of garnet toward lower *P-T* conditions on the decompression path (Fig. 1c). As a result, it postpones the onset of fracturing by delaying the moment at which the brittle field is reached. Consequently, a decrease in decompression rate during the decompression would prevent the formation of radial fractures if they are not already formed earlier along the decompression path. Similarly, a slow decompression at such high *T* would also lead to an efficient viscoelastic relaxation of the *P* in the inclusions, hence preventing any fracturing from occurring. Therefore, it is essential that the first stage of decompression occurs under fast decompression rates ≥8.0 GPa/Myr in order to predict the petrographic observations. Traditionally, the decompression path is related to the spatial displacement of geological units and exhumation rates (e.g., cm/yr). However, as suggested by Yamato and Brun[30], a fast decompression rate can be explained by a sudden change in the state of stress from compression to extension within the lithosphere. The substantial decompression rates of 8 GPa/Myr obtained from the thermo-mechanical modeling support this hypothesis because of the lack of solid evidence for such an enormous spatial (vertical or oblique) displacement[7].

The ability to model the visco-elasto-plastic rheology of garnet reveals important insights into the link between phase transition and deformation in inclusion-host systems. Here, the density change associated with the SiO$_2$ phase transition in inclusions of complex geometries has the potential to develop considerable shear stress, leading to fracturing of the garnet upon fast decompression. The results confirm that under the published range of decompression rates for the first stage of the decompression of the UHP Dora Maira Massif, garnet is fracturing at still high *P-T* conditions of >2 GPa and >600 °C, unlike previously thought based on elastic models. The major implication is that the onset of the decompression must be associated with a rapid switch in the stress state of the buried continental lithosphere, favoring the orogenic wedge model, in which nappe stacking triggers transient stress variations[26].

## Methods
### Code description
We use the 2D numerical code MDOODZ[38,42] based on the finite difference method/marker-in-cell approach, which is suitable for the study deformation of heterogenous visco-elasto-plastic materials. The

latest developments incorporate the effect of compressibility and progressive mineral transformations on rheology and strain localization[37]. The equation of momentum is formulated as:

$$\frac{\partial \tau_{ij}}{\partial x_i} - \frac{\partial P}{\partial x_i} = 0 \tag{1}$$

where $\tau_{ij}$ denotes the deviatoric strain rate tensor components, *P* is the pressure and $x_i$ corresponds to the spatial coordinates. Deviatoric stress is calculated using a Maxwell, visco-elasto-viscoplastic rheological model (e.g., ref. 43):

$$\dot{\varepsilon}_{ij} = \dot{\varepsilon}_{ij}^{e} + \dot{\varepsilon}_{ij}^{v} + \dot{\varepsilon}_{ij}^{vp} \tag{2}$$

where $\dot{\varepsilon}_{ij}$ corresponds to the deviatoric strain rate tensor components, and the e, v and vp superscripts respectively correspond to elastic, viscous and viscoplastic. The deviatoric elastic strain rate tensor is assumed to relate linearly with respect to the objective deviatoric stress rate:

$$\dot{\varepsilon}_{ij}^{e} = \frac{d\tau_{ij}}{dt} \tag{3}$$

Viscous creep strain rate follows a power law relation:

$$\dot{\varepsilon}_{ij}^{v} = 2^{-n} A e^{-\frac{E+PV}{RT}} \tau_{II}^{n-1} \tau_{ij} \tag{4}$$

where *A* is a material parameter, *E* is the activation energy, *V* is the activation volume, $\tau_{II}$ is the second deviatoric stress invariant and *n* is the power law index.

We assume that garnet yielding can be modeled using a pressure-dependent and dilatant frictional plastic rheological model. The emergence of shear failure patterns is hence modeled using a visco-plastic Drucker-Prager model. The yield (*F*) and potential (*Q*) functions are expressed as:

$$F = \tau_{II} - C \cos\varphi - P \sin\varphi - \dot{\lambda}\eta^{vp} \tag{5}$$

$$Q = \tau_{II} - P \sin\psi \tag{6}$$

where $\varphi$ is the friction angle, $\psi$ is the dilation angle, $C$ is the cohesion and $\eta^{vp}$ is the Kelvin element viscosity[44]. This model includes temporal regularization, which allows for successful satisfaction of force equilibrium at each time step. The viscoplastic plastic strain rate is expressed as:

$$\dot{\varepsilon}_{ij}^{vp} = \dot{\lambda}\frac{\partial Q}{\partial \tau_{ij}} \tag{7}$$

where $\dot{\lambda}$ is the rate of the plastic multiplier, which is determined using an analytical return mapping procedure based on trial yield function and material parameters (see details in ref. [42]). Frictional shear banding only occurs if the pressure-dependent yield stress is attained. At such temperatures (>600 °C), this limit can be reached in strong hosts like garnet but not in weaker (e.g., omphacite, Supplementary Fig. 2).

The model is compressible and the continuity equation is expressed as:

$$\frac{d \ln(\rho)}{dt} = -\frac{\partial v_i}{\partial x_i} \tag{8}$$

The simulations thus account for volumetric strains induced by elasto-plastic deformations and by phase transitions. The density ($\rho$) is function of pressure and temperature. For phases that involve phase transitions (i.e., $SiO_2$), the equilibrium density ($\rho$) is pre-computed using Gibbs free energy minimization and stored in look-up tables. For other materials, the following equation of state is applied:

$$\rho = \rho_0 e^{\beta P - \alpha T} \tag{9}$$

where $\rho_0$ is the reference density, $\beta$ is the incompressibility modulus and $\alpha$ is the thermal expansivity. Here, the use of a constant compressibility for garnet is a simplifying assumption[45]. We have tested several values of compressibility ($K = \beta^{-1}$). These variations did not have any significant effect (Supplementary Fig. 3), which is also in line with a recent study by Moulas et al. [46].

In case of phase transitions, the density evolves as:

$$\frac{d\rho}{dt} = \frac{\rho^{eq} - \rho}{t^k} \tag{10}$$

where $t^k$ is the transformation kinetic time, which is expressed as:

$$t^k = \ln\left(\frac{1}{3}\right)\left(-2S\dot{g}\right)^{-1} \tag{11}$$

where $S$ is the grain boundary area. The growth rate is:

$$\dot{g} = k_0 T e^{-\frac{Q_g}{RT}}\left(1 - e^{-\frac{\Delta G}{RT}}\right) \tag{12}$$

where $\triangle G$ is the free energy change of reaction computed using Gibbs free energy minimization, $Q_g$ is the activation energy for growth and $k_0$ is a constant[39]. Note that at the considered temperature, growth kinetic rates are much larger than considered decompression rates. Kinetics has thus a negligible effect as transformation rates are quasi-instantaneous.

## Model configuration

The model configuration is characterized by a 2D medium of $5 \times 5$ mm, consisting of a matrix $SiO_2$, containing a garnet crystal with several $SiO_2$ inclusions (Fig. 2b). The geometry is based on the digitalized thin section microphotography in Fig. 2a. The density-pressure relationship of $SiO_2$ is calculated based on the Holland and Powell 2011 database[47]

and the density evolution of garnet follows the equation of state. Note that the variation of the molecular volume of $\alpha$-quartz in Holland and Powell 2011[47] is equal to the one in their 1998 database[48], which does not significantly differ from other estimates in the literature within the temperature range of interest (730–600 °C)[49]. The kinetics of the coesite to quartz phase transition is taken into account in the models following Mosenfelder and Bohlen[39] ($Q_g = 269 \pm 26$ kJ/mol, $k_0 = 26.537$ m/s/K). Creep parameters are based on laboratory-derived flow laws: garnet creep follows the flow law of Ji and Martignole[50], and $SiO_2$ creep was modeled using dry quartz experimental data[51,52] (see Supplementary Table 1 for material parameters).

The initial conditions are set to the peak Alpine P-T conditions of 4.3 GP and 730 °C, according to Hermann[16]. The decompression of the model from the peak conditions follows the retrograde P-T path constrained by the peak and retrograde P-T data from Hermann[16] and Rubatto and Hermann[4], as shown in Fig. 1a. The temperature (unit Kelvin) is assumed constant at the scale of the sample and is updated at each time step using the following regression of the P-T data:

$$T = 273.15 + \frac{1}{a}\log_{10}\left(\frac{P \times 10^{-9}}{b}\right) \tag{13}$$

where $a = 1.02 \times 10^{-2}$ and $b = 2.523 \times 10^{-3}$.

Progressive dilation and associated decompression are applied via kinematic boundary conditions. The velocity component normal to each boundary ($v_i$) is updated at each time increment as:

$$v_i = x_i \times \nabla v^{BG} \tag{14}$$

where $x_i$ is the current boundary coordinate and $\nabla v^{BG}$ is the applied dilation rate, which is assumed constant with time. For materials involving no phase transition, the decompression rate can be directly related to the dilation rate. However, for materials involving a phase transition, the pressure remains buffered until the phase transition is complete. For this reason, the modeled decompression rates vary with time, although the dilation rate is constant (see Fig. 4a). The provided decompression rates are thus averaged over the decompression paths. We model purely isotopic dilation of the sample, and we ensure that boundary conditions do not induce deviatoric strains. To this end, the following non-zero out-of-plane strain rate is applied:

$$\frac{\partial v_y}{\partial y} = \frac{1}{2}\left(\frac{\partial v_x}{\partial x} + \frac{\partial v_z}{\partial z}\right) \tag{15}$$

All internally modeled deviatoric strains (e.g., shear bands) are thus solely induced by stress concentrations and caused by internal sample heterogeneity. The development of shear bands could be further enhanced if tectonic deviatoric deformation fields were additionally considered (e.g., pure or simple shear). Our models thus provide conservative estimates on the development of shear failure. In order to further verify our 2D models, we have run 1D host-inclusion-matrix decompression models with radial symmetry (Supplementary Fig. 4). We have tested two types of boundary conditions: (1) constant dilation rate and (2) constant decompression rates. Both types of models show that fast decompression rates (>5 GPa/Myr) or fast dilation rates (>$5 \times 10^{-15}$ s$^{-1}$) are needed to reach frictional yielding of garnet, thus confirming the presented 2D results.

## Data availability

The presented results are based on numerical simulations and do not involve source data.

## Code availability

The data generated in this study have been performed using the code MDoodz7.0 [https://zenodo.org/record/8112302]. This code is freely

accessible. All the information needed to install and use it, as well as any updates, can be found here: https://github.com/tduretz/MDOODZ7.0.

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

## Acknowledgements

This work was supported by the Goethe University of Frankfurt, an Alexander von Humboldt research grant to C.L. and a Wilhelm and Else Heraeus Foundation visiting professorship to P.Y. The Heidelberg preparation lab is acknowledged for high-quality thin-section preparation. We thank Taras Gerya for his detailed and constructive reviews.

## Author contributions

All authors participated equally in the design of the study. C.L.: Formal analysis, investigations, numerical simulations & visualization, writing—original draft & editing; L.T.: Writing original draft & editing and revision; P.Y.: Involvement in the numerical code development, writing—editing and revision; T.D.: Numerical code development, writing—editing and revision, supervision.

## Competing interests

The authors declare no competing interests.
