## [Peer Review File · Nature Communications]

REVIEWER COMMENTS

Reviewer #1 (Remarks to the Author):

This is an interesting and provocative paper estimating for the first time very high decompression rates of untrahigh-pressure rocks based on combination of natural data with results of advanced numerical thermomechanical modeling. The paper is of broad interest and suggest that decompression of these rocks may rather be predominantly related to rapid stress changes under orogens and not to physical movement of untrahigh-pressure rocks toward the surface (exhumation). However, the paper misses some important details on the used modeling approach and the numerical model design that needs to be added to Methods (see specific comments). Also, decompression rates are characterized indirectly by dilatation rate ($1/s$), which is confusing and needs improvement. In addition, the used equation of state for alpha-quartz needs careful check since its molecular volume behavior is rather complex in the studied PT region and the used model of Holland and Powell (2011) may be suboptimal. Specific comments are given below.

Specific comments

Line 22. "Plate tectonics is a key driver of natural phenomena occurring on Earth, such as the emergence of life, global element cycles, climate evolution and natural disasters. How plate tectonics has evolved through time is still one of the fundamental questions in Earth sciences^{1–3}. Natural microstructures observed in exhumed ultrahigh-pressure rocks formed during continental collision provide crucial insights into tectonic processes in the Earth's interior." Plate tectonic may not necessarily be related to life emergence but rather to its evolution. Also, there is no explicit connection between the sentences. Better to modify the first one as something like "Plate tectonics is a key driver of many natural phenomena occurring on Earth, such as the movements and collision of continents, mountain building and erosion, global element cycles, climate evolution, natural disasters and the evolution of life."

Line 31. «The decompression rates higher than 10-14 s⁻¹ ...» The units correspond to velocity divergence ($1/s$) rather than to decompression rate (Pa/s). Better to also add and discuss more intuitive quantities (decompression rate, kbar/Myr, inferred exhumation rate, km/Myr).

Line 35. "Our results question the traditional interpretation of fast and significant vertical displacement of ultrahigh-pressure tectonic units during exhumation. Instead, we propose that such ultrafast decompression rates are related to abrupt changes in the stress state of the lithosphere." Here is a good place to interpret the proposed decompression rate in term of needed exhumation rate (see above).

Line 28. “The iconic petrographic observations in the UHP rocks are the partial preservation of UHP coesite inclusions in garnet that were partially to completely turned into quartz - the low pressure SiO₂ polymorph - during retrogression, and the associated radial fractures in the garnet host.” Would be good to specify that it was the transition to alpha-quartz.

Line 173. “For a decompression rate of 10-15 s⁻¹, diffuse plastic deformation occurs at a late stage (~ 1.5 GPa), once all SiO₂ turned to quartz. If the decompression rate is lower than 10-15 s⁻¹, garnet behaves in a ductile manner from the onset of exhumation until ca. 2 GPa.” Again, better to use more intuitive measure for decompression rate (see above).

Line 206. “The decompression rates based on geochronological data of Dora Maira⁴ between the peak conditions of 4.3 GPa and the retrograde conditions of 1.1 GPa translate into constant decompression rates of 2.3×10⁻¹⁴ s⁻¹ (equivalent to a minimum duration of ~ 0.4 Ma) to 3.5×10⁻¹⁵ s⁻¹ (equivalent to a maximum duration of ~ 4 Ma), for which fracturing is enabled (Fig.4b).” Same as above.

Line 441. “The application of background bulk decompression rate at the model’s boundary allows for simulating a progressive decompression. The kinetics of the coesite to quartz phase transition is taken into account in the models following Mosenfelder and Bohlen³⁹.” More details on the modeling approach are needed. It is important to discuss which equations were eventually solved. Was it plain-strain approximation? Supplementary figure with used boundary conditions is needed. How decompression was modeled? Was it applied as time-dependent pressure boundary condition (where, how?) or as a constant outward velocity of the material at the boundaries. What about temperature changes? Were they prescribed as a function of time or something else?

Line 434. “The density-pressure relationship of SiO₂ is calculated based on the Holland and Powell 2011 database⁴³ and the density evolution of garnet follows the equation of state (EOS).” Since alpha-quartz has large and complex molecular volume (and density) variations in the studied P-T region (e.g., Gerya et al., 2004 and references therein) it would be good to double check if these volume changes are correctly reproduced by the Holland and Powell 2011 database – it seems to rely on Holland and Powell, 1998 EOS for quartz, which strongly underestimates alpha-quartz molecular volume changes (Gerya et al., 2004, Fig. 5b).

References

Gerya T.V., Podlesskii K.K., Perchuk L.L., Maresch, W.V. (2004) Semi-empirical Gibbs free energy formulations for minerals and fluids. *Phys. Chem. Minerals*, 31, 429-455.

Reviewer #2 (Remarks to the Author):

I appreciate the opportunity to review the manuscript by Luisier et al. focusing on the microstructure of a UHP garnet from Dora Maira whiteschists. Conventional wisdom links the radial cracking around silica inclusion to brittle fracturing in response to differential volumetric expansion, which takes place at relatively low pressures. This study employs a visco-elastic model and shows that the plastic shear bands appear similar to the fracture pattern. The decompression rate of the first stage exhumation would have been fast.

Although I am convinced in principle, I have a few concerns that the authors might address or clarify in the revised manuscript. I hope my comments could be helpful.

1) Radial cracks in garnet are almost ubiquitous in UHP rocks. Do they all suggest very fast decompression? This is an issue of sufficiency vs. necessity of the argument. I understand that the authors are not suggesting fast decompression of all UHP rocks (although they cite Yamato and Brun, 2017). The peak temperature of Dora Maira, as well as the initial exhumation P-T path, happens to be close to the 10^{-14} s⁻¹ contour of garnet brittle-ductile transition. The converted decompression rate of Dora Maira is also close to 10^{-14} . The coincidences, with no geodynamic connections, result in the specific fracturing pattern of the garnet from the Dora Maira whiteschist. In other cases, if the peak-T of UHP metamorphism is lower (e.g., < 650 °C) or the exhumation is slower (note that 5 cm/yr is very fast), garnet behaves more elastically, so this model does not apply. I suggest some discussion added to the “Global significance” section to guard this paper against being overinterpreted. I also note that “global significance” might sound misleading.

2) I had a hard time finding how the pattern in Fig. 2c is similar to Fig. 2a. I am not questioning the authors’ statement, but I simply did not know what to look at. The radial and parallel cracks in Fig. 2a are not clear to me, either. Maybe the authors could use different colors to highlight the radial and parallel cracks in Fig. 2b or another panel. Please also briefly describe the similarities in the text and guide the readers through, instead of forcing the readers to accept the statement (Line 107-108).

3) In the text, the authors unnecessarily create confusion about “decompression strain rate” and “decompression rate”. I believe the strain rate is what the authors meant, but it is not defined (vaguely mentioned in the caption of Fig. 2). The strain rate boundary condition needs a better explanation and justification. For example, Line 186-188 could be moved forward, and please explain how the exhumation rate (GPa/myr) is translated to the strain rate (s⁻¹). More importantly, please consider these four points:

- The BDT transition strain rate of garnet (contours in Fig. 1c) and the strain rate of the model space (Fig. 2b) are conceptually irrelevant, but the text and Fig. 3 gave me the wrong impression that they are the same. I understand that Figure 1 presents the underlying motivation.

- How are the relative sizes of garnet and matrix (Fig. 2b) determined, and how does the variation of the model space affect the conclusions? I guess the fracturing behavior depends on the relative sizes (volumes) of garnet and the matrix, as they have different bulk moduli and rheological properties?

- At a constant strain rate 10^{-14} s^{-1} , for 300 kyr (Fig. 4a) and 1 myr (Fig. 1a), the linear expansion is about 10% and 30%, respectively. Do the numbers appear too big?

- The relaxation for 200 kyr at the coesite-quartz transition (Fig. 4a) is a physical artifact of the constant-strain-rate boundary condition, but does not make sense geologically. The exhumation of a UHP rock in a subduction zone would not stall at a specific pressure (or depth if lithostatic). Maybe the authors would consider another model with real volume strain rates along the exhumation P-T-t path, which is a function of the volume expansion in response to P, T, and phase transition.

4) The fracturing behavior of garnet critically depends on the thermos-mechanical parameters of garnet and quartz, so the models used and their uncertainties deserve further discussion, in addition to Lines 437-439. For example, I am curious about the authors' choice—why use Holland and Powell (2011) for quartz but a different set of numbers for garnet? Do we need to consider the dependence of these parameters (K and α) on P-T, and the flow laws on other factors like the water content? I presume the model results would not change much when these parameters vary in their reasonable ranges, but a sensitivity test that does not involve much additional work would make the manuscript more robust.

In addition, beyond the scope of this study, coesite/quartz inclusions in omphacite and kyanite typically result in minor or no radial cracks. Would the authors throw in the thermo-mechanical parameters of these minerals (ignoring cleavage and anisotropy first) and see if they crack differently? If they also produce similar shear bands as in Fig. 3, the model and the conclusion from this manuscript should be more carefully examined.

Some minor comments:

Line 25-26: "significant role in ..." is vague. "UHP rocks - ... - are the key witness of ..." reads better, in my opinion.

Line 34: Are these 14 references all necessary?

Line 39: Many readers are not experts on UHP metamorphism or subduction zone geodynamics. The authors might comment a bit on the significance of this exhumation rate.

Line 85-88: See my comment #1. Please clarify if this statement is specific to Dora Maira.

Line 176-177: Yes, I have the same question when I read this manuscript. If the authors have speculations, it does not hurt to say them explicitly.

Line 189: Please be more specific. How early are the fractures formed?

Line 202-204: My impression is that the switch of stress state in Yamato & Brun's model should be faster than hundreds of kyrs, but I could be wrong. I do not think Yamato & Brun (2017) has been a consensus in the community. The model is inconsistent with "our knowledge of the burial and exhumation cycle" (Line 33-34 and similar language elsewhere).

Line 217-221: This sentence is too general and does not appear to add much to the text. Instead, I'd be more interested in the insights or comments on how the model presented could be tested in future studies because the model and results presented in this manuscript are more like a hypothesis.

Figure 4b: Are the crosses missed on the green line, or does this line mean something different?

Line 406: I am not a modeler, so I do not understand what "large" means in this context.

Line 411: How would the model results change with a smaller or larger model space?

Line 415: Please give references here and for the numbers in Table 1. Why do the authors use a different EOS from Holland and Powell (2011) for garnet?

Line 422-423: Please briefly describe the kinetic model, or provide the equation.

REVIEWER COMMENTS

Reviewer #1 (Remarks to the Author):

This is an interesting and provocative paper estimating for the first time very high decompression rates of untrahigh-pressure rocks based on combination of natural data with results of advanced numerical thermomechanical modeling. The paper is of broad interest and suggest that decompression of these rocks may rather be predominantly related to rapid stress changes under orogens and not to physical movement of untrahigh-pressure rocks toward the surface (exhumation). However, the paper misses some important details on the used modeling approach and the numerical model design that needs to be added to Methods (see specific comments). Also, decompression rates are characterized indirectly by dilatation rate (1/s), which is confusing and needs improvement. In addition, the used equation of state for alpha-quartz needs careful check since its molecular volume behavior is rather complex in the studied PT region and the used model of Holland and Powell (2011) may be suboptimal. Specific comments are given below.

We are pleased that the reviewer#1 found this manuscript interesting and important! Thank you! We tackle the specific comments below.

Specific comments

Line 22. "Plate tectonics is a key driver of natural phenomena occurring on Earth, such as the emergence of life, global element cycles, climate evolution and natural disasters. How plate tectonics has evolved through time is still one of the fundamental questions in Earth sciences1–3. Natural microstructures observed in exhumed ultrahigh-pressure rocks formed during continental collision provide crucial insights into tectonic processes in the Earth's interior." Plate tectonic may not necessarily be related to life emergence but rather to its evolution. Also, there is no explicit connection between the sentences. Better to modify the first one as something like "Plate tectonics is a key driver of many natural phenomena occurring on Earth, such as the movements and collision of continents, mountain building and erosion, global element cycles, climate evolution, natural disasters and the evolution of life."

Thank you for the suggestion. We modified the text accordingly.

Line 31. «The decompression rates higher than 10-14 s⁻¹ ...» The units correspond to velocity divergence (1/s) rather than to decompression rate (Pa/s). Better to also add and discuss more intuitive quantities (decompression rate, kbar/Myr, inferred exhumation rate, km/Myr).

Thank you for pointing this out. This is a good point, indeed! We modified the text accordingly. We now use the term "decompression rate" with appropriate units (GPa/Myr), which corresponds to at least 8 GPa/Myr for the studied microstructures. This conversion from the dilation rate to decompression rate is described in the new Fig. 4c and in the text. A specific section dedicated to this is also provided in the Methods (see l. 534-546).

Interestingly, the decompression rate agrees with the current petrochronological data as indicated in lines 196-201, and we question the traditional conversion of the petrochronological data to depth because the decompression is large and ultrafast.

Line 35. "Our results question the traditional interpretation of fast and significant vertical displacement of ultrahigh-pressure tectonic units during exhumation. Instead, we propose that such ultrafast decompression rates are related to abrupt changes in the stress state of the lithosphere." Here is a good place to interpret the proposed decompression rate in term of needed exhumation rate (see above).

This is related to our answer above. Following the suggestions of reviewer#1, we modified the text accordingly. We just want to highlight that our model does not predict spatial displacement.

Line 28. "The iconic petrographic observations in the UHP rocks are the partial preservation of UHP coesite inclusions in garnet that were partially to completely turned into quartz - the low pressure SiO₂ polymorph - during retrogression, and the associated radial fractures in the garnet host." Would be good to specify that it was the transition to alpha-quartz.

We modified the "quartz" to "α-quartz", where appropriate.

Line 173. "For a decompression rate of 10-15 s⁻¹, diffuse plastic deformation occurs at a late stage (~ 1.5 GPa), once all SiO₂ turned to quartz. If the decompression rate is lower than 10-15 s⁻¹, garnet behaves in a ductile manner from the onset of exhumation until ca. 2 GPa." Again, better to use more intuitive measure for decompression rate (see above).

As above, following the suggestions of reviewer#1, we modified the text accordingly. We now use the decompression rate (GPa/Myr) or dilation rate (s^{-1}).

Line 206. "The decompression rates based on geochronological data of Dora Maira⁴ between the peak conditions of 4.3 GPa and the retrograde conditions of 1.1 GPa translate into constant decompression rates of $2.3 \times 10^{-14} s^{-1}$ (equivalent to a minimum duration of ~ 0.4 Ma) to $3.5 \times 10^{15} s^{-1}$ (equivalent to a maximum duration of ~ 4 Ma), for which fracturing is enabled (Fig.4b)." Same as above.

Same as above. The "decompression rate" is now in GPa/Myr.

Line 441. "The application of background bulk decompression rate at the model's boundary allows for simulating a progressive decompression. The kinetics of the coesite to quartz phase transition is taken into account in the models following Mosenfelder and Bohlen³⁹." More details on the modeling approach are needed. It is important to discuss which equations were eventually solved. Was it plain-strain approximation? Supplementary figure with used boundary conditions is needed. How decompression was modeled? Was it applied as time-dependent pressure boundary condition (where, how?) or as a constant outward velocity of the material at the boundaries. What about temperature changes? Were they prescribed as a function of time or something else?

The Methods part was modified. We added the requested details in this chapter. The boundary conditions were tested and the results are in Supplementary information (Fig. S4).

Line 434. "The density-pressure relationship of SiO₂ is calculated based on the Holland and Powell 2011 database⁴³ and the density evolution of garnet follows the equation of state (EOS)." Since alpha-quartz has large and complex molecular volume (and density) variations in the studied P-T region (e.g., Gerya et al., 2004 and references therein) it would be good to double check if these volume changes are correctly reproduced by the Holland and Powell 2011 database – it seems to rely on Holland and Powell, 1998 EOS for quartz, which strongly underestimates alpha-quartz molecular volume changes (Gerya et al., 2004, Fig. 5b).

The molecular volume of alpha-quartz in Holland and Powell 2011 is the same as the one in the 1998 database (H&P98). The evolution of the molar volume versus temperature of alpha-quartz in H&P98 differs from the curve in Fig. 5b of Gerya et al., (2004) at temperatures lower than 800 K. Above this temperature, all curves are overlapping. In our study, we start the models at the peak conditions of 1003 K (730 °C) and decrease to the temperature of 873 K (600 °C), corresponding to the retrograde stage of 1.1 GPa. Hence, we are within the temperature range for which the molecular volume of alpha-quartz is similar between the H&P database and Gerya et al., 2004. See Figure below, showing the comparison of the coe/alpha-qtz using the two Holland and Powell databases:

This point is also now specified in the Methods (see l. 517-522)

References

Gerya T.V., Podlesskii K.K., Perchuk L.L., Maresch, W.V. (2004) Semi-empirical Gibbs free energy formulations for minerals and fluids. *Phys. Chem. Minerals*, 31, 429-455.

Reviewer #2 (Remarks to the Author):

I appreciate the opportunity to review the manuscript by Luisier et al. focusing on the microstructure of a U}TP garnet from Dora Maira whiteschists. Conventional wisdom links the radial cracking around silica inclusion to brittle fracturing in response to differential volumetric expansion, which takes place at relatively low pressures. This study employs a visco-elastic model and shows that the plastic shear bands appear similar to the fracture pattern. The decompression rate of the first stage exhumation would have been fast.

Although I am convinced in principle, I have a few concerns that the authors might address or clarify in the revised manuscript. I hope my comments could be helpful.

We thank reviewer#2 for reading our manuscript and providing the helpful comments. We tackle all of them below.

1) Radial cracks in garnet are almost ubiquitous in U}TP rocks. Do they all suggest very fast decompression? This is an issue of sufficiency vs. necessity of the argument. I understand that the authors are not suggesting fast decompression of all U}TP rocks (although they cite Yamato and Brun, 2017). The peak temperature of Dora Maira, as well as the initial exhumation P-T path, happens to be close to the 10^{-14} s $^{-1}$ contour of garnet brittle-ductile transition. The converted decompression rate of Dora Maira is also close to 10^{-14} . The coincidences, with no geodynamic connections, result in the specific fracturing pattern of the garnet from the Dora Maira whiteschist. In other cases, if the peak-T of U}TP metamorphism is lower (e.g., < 650 °C) or the exhumation is slower (note that 5 cm/yr is very fast), garnet behaves more elastically, so this model does not apply. I suggest some discussion added to the "Global significance" section to guard this paper against being overinterpreted. I also note that "global significance" might sound misleading.

This comment might be related to the confusion about the decompression rate vs dilation rate similarly mentioned by the reviewer#1, based on the early version of the manuscript. We modified and improved the text accordingly (see Methods and text related to Fig. 4).

The decompression rate is not 10^{-14} , therefore, this must be a misunderstanding. 10^{-14} s $^{-1}$ is the dilation rate. We now hope that after the overall modification, it is clear.

The constraint for Dora Maira is that, it is at high-temperature in order to be sure that the rate is large enough to induce plasticity. In other words, the low decompression rate is connected with viscous behavior. Therefore, we need high decompression rate to fit the observations.

2) I had a hard time finding how the pattern in Fig. 2c is similar to Fig. 2a. I am not questioning the authors "statement, but I simply did not know what to look at. The radial and parallel cracks in Fig. 2a are not clear to me, either. Maybe the authors could use different colors to highlight the radial and parallel cracks in Fig. 2b or another panel. Please also briefly describe the similarities in the text and guide the readers through, instead of forcing the readers to accept the statement (Line 107-108).

The dilatant plastic shear bands in Fig 2c appear in orange on the figure. We chose this colormap on purpose, to highlight the deformation pattern. We describe this pattern in the main text (lines 111-113) and their condition of occurrence in the new method section, line 473-486.

3) In the text, the authors unnecessarily create confusion about "decompression strain rate" and "decompression rate". I believe the strain rate is what the authors meant, but it is not defined (vaguely mentioned in the caption of Fig. 2). The strain rate boundary condition needs a better explanation and justification. For example, Line 186-188 could be moved forward, and please explain how the exhumation rate (GPa/myr) is translated to the strain rate (s^{-1}). More importantly, please consider these four points:

This should be now clarified also based on comments of reviewer#1. Decompression rate (in GPa/Myr) is now used through the manuscript.

- The BDT transition strain rate of garnet (contours in Fig. 1c) and the strain rate of the model space (Fig. 2b) are conceptually irrelevant, but the text and Fig. 3 gave me the wrong impression that they are the same. I understand that Figure 1 presents the underlying motivation.

It is not clear to us what exactly the reviewer #2 means by this comment. The Fig. 3 is based on Fig 1c result, i.e. the Fig. 3 takes the given BDT line that is relevant for our case. We do not understand the reviewer#2' relation to Fig. 2b. The Fig. 2b is not related to the BDT arguments.

- How are the relative sizes of garnet and matrix (Fig. 2b) determined, and how does the variation of the model space affect the conclusions? I guess the fracturing behavior depends on the relative sizes (volumes) of garnet and the matrix, as they have different bulk moduli and rheological properties?

This is an excellent point. Thank you for this. It is now documented in the new Supplementary information (Fig. S1), where the relation between "how much matrix" vs. garnet and the fracturing behavior is investigated. Results show that varying the matrix proportion did not impact the pressure at which yielding initiated.

- At a constant strain rate 10^{-14} s^{-1} , for 300 kyr (Fig. 4a) and 1 myr (Fig. 1a), the linear expansion is about 10% and 30%, respectively. Do the numbers appear too big?

One third of the dilation rate is applied along each spatial direction (see new method section). Hence the estimated 10% appears reasonable. In practice, the expansion is partitioned between the phases as they are characterized by different mechanical properties (compressibility, occurrence or not of phase transition).

- The relaxation for 200 kyr at the coesite-quartz transition (Fig. 4a) is a physical artifact of the constant-strain-rate boundary condition, but does not make sense geologically. The exhumation of a UHP rock in a subduction zone would not stall at a specific pressure (or depth if lithostatic). Maybe the authors would consider another model with real volume strain rates along the exhumation P-T-t path, which is a function of the volume expansion in response to P, T, and phase transition.

To tackle this comment, we tested various boundary conditions and scenarios that are now provided in Supplementary information S4.

Furthermore, as indicated above, our model does not provide spatial displacement, thus exhumation rates. We only provide decompression rates directly comparable with P-T-t data, which is now clarified in the text.

4) The fracturing behavior of garnet critically depends on the thermo-mechanical parameters of garnet and quartz, so the models used and their uncertainties deserve further discussion, in

addition to Lines 437-439. For example, I am curious about the authors "choice—why use Holland and Powell (2011) for quartz but a different set of numbers of garnet? Do we need to consider the dependence of these parameters (K and α) on P-T, and the flow laws on other factors like the water content? I presume the model results would not change much when these parameters vary in their reasonable ranges, but a sensitivity test that does not involve much additional work would make the manuscript more robust.

This is now described in detail in the Methods section. Additional models including sensitivity to boundary loading type, host creep rheology, host compressibility and matrix amount are now included in the Supplementary Material. They do not impact much the model results but do indeed make the study more robust.

In addition, beyond the scope of this study, coesite/quartz inclusions in omphacite and kyanite typically result in minor or no radial cracks. Would the authors throw in the thermo-mechanical parameters of these minerals (ignoring cleavage and anisotropy first) and see if they crack differently? If they also produce similar shear bands as in Fig. 3, the model and the conclusion from this manuscript should be more carefully examined.

Following your advice, we run the simulation for omphacite as well and now it is as Supplementary figure S2. An omphacite creep law does not allow to build sufficient deviatoric stress and does not trigger frictional plasticity of the host, despite the phase transitions that occurred in both the matrix and inclusion.

Some minor comments:

Line 25-26: "significant role in ..." is vague. "UHP rocks - ... - are the key witness of ..." reads better, in my opinion.

The text was modified accordingly.

Line 34: Are these 14 references all necessary?

Besides we used all these data to constrain the decompression path, the aim is to show that this area is very well studied and all the P-T estimates agree more or less, i.e., the data that we use to constrain the decompression path are robust.

Line 39: Many readers are not experts on UHP metamorphism or subduction zone geodynamics. The authors might comment a bit on the significance of this exhumation rate.

As suggested, the significance of the exhumation rate and how it is calculated is now clarified in the text (see l. ~39-42).

Line 85-88: See my comment #1. Please clarify if this statement is specific to Dora Maira.

We clarified as suggested.

Line 176-177: Yes, I have the same question when I read this manuscript. If the authors have speculations, it does not hurt to say them explicitly.

In lines 183-188, we added what we think could be responsible for the preservation of coesite, based on our results so far.

Line 189: Please be more specific. How early are the fractures formed?

The conditions at which the fracturing starts are discussed in lines 145-147 and 201-203. Furthermore, it is also mentioned in the second part of the sentence: " soon after the phase transition has occurred in the matrix and while the phase transition is still occurring within the inclusions."

Line 202-204: My impression is that the switch of stress state in Yamato & Brun's model should be faster than hundreds of kyrs, but I could be wrong. I do not think Yamato & Brun (2017) has been a consensus in the community. The model is inconsistent with "our knowledge of the burial and exhumation cycle" (Line 33-34 and similar language elsewhere).

This is exactly the reason why we think it is interesting to get decompression rates independent of petrochronological data. So far, the P-T-t data could not be used to discriminate different geodynamic models. Here, we have both, the P-T-t evolution as well as the microstructural evidence, as the key arguments for very fast decompression. Furthermore, we show with our results that the switch of stress can be much faster at the beginning of the decompression. This is a very important and provocative finding, documented for the first time.

Line 217-221: This sentence is too general and does not appear to add much to the text. Instead, I'd be more interested in the insights or comments on how the model presented could be tested in future studies because the model and results presented in this manuscript are more like a hypothesis.

We deleted this sentence/paragraph that connected our study beyond the geologically relevant context. Instead, we provided more tests on boundary conditions (Supplementary information) which are described in Methods.

Figure 4b: Are the crosses missed on the green line, or does this line mean something different?

No, it means that the plastic deformation is not similar to the results at higher decompression rates. Here, the plastic deformation is diffuse and does not form the patterns observed in the samples. We now clarify this point by adding dots in figure 4b.

Line 406: I am not a modeler, so I do not understand what "large" means in this context.

Large strain means that the model takes into account advection, rotation and deformation of the stress tensor. This technicality is not mentioned in the text anymore.

Line 411: How would the model results change with a smaller or larger model space?

We have tested it, and the results are now implemented in Fig. 4.

Line 415: Please give references here and for the numbers in Table 1. Why do the authors use a different EOS from Holland and Powell (2011) for garnet?

The Methods part was improved and more details are now provided concerning this point.

Line 422-423: Please briefly describe the kinetic model, or provide the equation.

A detailed description of the code and equations solved is now provided in the Methods part.

REVIEWERS' COMMENTS

Reviewer #1 (Remarks to the Author):

The authors did a good job in addressing comments and revising the paper. I found only one typo in Methods:

Line 479. It is written: "The equations of momentum and continuity are formulated as:". Should be: "The equation of momentum is formulated as:"

Reviewer #2 (Remarks to the Author):

I appreciate the author's diligent attention to the previous remarks of reviewers and for carrying out extensive revisions. The manuscript now offers greater accessibility due to improved sections on methodology and enhanced explanations scattered throughout the paper. The clarity of the model assumptions and parameters enables fellow researchers to form independent evaluations. This impressive study illustrates how much the petrographic textures can imply and how much insights a time dimension of "extreme condition" of metamorphism can provide. This paper is likely to spark a critical reassessment of coesite inclusions in UHP garnet, which could potentially revolutionize our comprehension of the subduction-exhumation processes of (U)HP terrains.

There are a few minor points that the authors might wish to address. The relationship between the "mean" decompression and dilation rates remains somewhat unclear. Could the authors clarify why this relationship is linear and explain the role of the x_i coefficient in the equation (Line 557)? Has the coe-qrt transition not been factored into this linear relationship? The authors employed a boundary condition of constant dilation rate, thus resulting in a decompression rate that changes over time. Is the decompression rate the one directly influenced by tectonics? For instance, the stagnation of decompression at the coe-qrt transition could be an artifact of the model setup. In reality, this polymorphic transition does not affect the tectonic shift in the stress state (or exhumation). It would be interesting to see the results of the model had a constant-decompression-rate boundary condition been applied. I assume this appears similar to the green curve in Fig. 4a (phase transition in the inclusion but not the matrix)? I am not suggesting the authors run another model, but some brief thoughts on this would be appreciated.

REVIEWER COMMENTS

Reviewer #1 (Remarks to the Author):

The authors did a good job in addressing comments and revising the paper. I found only one typo in Methods:

Line 479. It is written: “The equations of momentum and continuity are formulated as:”. Should be: “The equation of momentum is formulated as:”

We modified this sentence as requested

Reviewer #2 (Remarks to the Author):

I appreciate the author’s diligent attention to the previous remarks of reviewers and for carrying out extensive revisions. The manuscript now offers greater accessibility due to improved sections on methodology and enhanced explanations scattered throughout the paper. The clarity of the model assumptions and parameters enables fellow researchers to form independent evaluations. This impressive study illustrates how much the petrographic textures can imply and how much insights a time dimension of “extreme condition” of metamorphism can provide. This paper is likely to spark a critical reassessment of coesite inclusions in UHP garnet, which could potentially revolutionize our comprehension of the subduction-exhumation processes of (U)HP terrains.

There are a few minor points that the authors might wish to address. The relationship between the "mean" decompression and dilation rates remains somewhat unclear. Could the authors clarify why this relationship is linear and explain the role of the x_i coefficient in the equation (Line 557)?

For models with kinematic boundary conditions, the pressure close the boundary is the result of the computation. In the models driven by dynamic conditions (decompression rate), the boundary velocity (and thus resulting dilation rate) is a result of the computation. For the 2D models, we have assumed kinematic boundary conditions and the boundary divergence rate is constant with time. The mean matrix decompression rate, probed close to the boundaries, is not constant with time. The boundary velocity is set to:

$$v_i = x_i \times \nabla v^{BG}$$

where v_i is velocity of the boundary, x_i is the coordinate of the boundary (updated at each time increment) and the ∇v^{BG} is the dilation rate applied at the boundary. There is no assumption of a linear relationship between dilation rate and decompression rate, the latter is a result of the numerical solution.

Has the coe-qrt transition not been factored into this linear relationship?

No, the coe-q transition is not directly related to this equation. It is implemented by including the effect of density variations (function of both P and T) in the continuity equation.

The authors employed a boundary condition of constant dilation rate, thus resulting in a decompression rate that changes over time. Is the decompression rate the one directly influenced by tectonics? For instance, the stagnation of decompression at the coe-qrt transition could be an artifact of the model setup.

There are two possibilities, one can assume either kinematic or dynamic boundary conditions.

1. *The kinematic boundary conditions (used in 1D and 2D models) assumes that the system is governed by far-field kinematics (i.e. tectonics). In that case, pressure buffering during phase transition is a natural consequence of the applied constant dilation rate.*

2. *The dynamic boundary condition assumes that the system is governed by surrounding decompression rate. Large transient variations of the sample divergence rate are a natural consequence of that choice.*

To our knowledge, there is no reason to favor one hypothesis over the other. Both choices can be made equally and their consequences were explained during the first revision round. The most important here is that frictional plastic yielding of garnet either implies or require large decompression rates.

In reality, this polymorphic transition does not affect the tectonic shift in the stress state (or exhumation). It would be interesting to see the results of the model had a constant-decompression-rate boundary condition been applied. I assume this appears similar to the green curve in Fig. 4a (phase transition in the inclusion but not the matrix)? I am not suggesting the authors run another model, but some brief thoughts on this would be appreciated.

We have actually presented results assuming constant boundary decompression rate after the first revision (1D only). Indeed, under this type of (un)loading, the boundary pressure decreases steadily with time as it is constrained to behave this way. It then looks like the green curve of Fig.4A. However, the dilation rate experiences transient variations required to accommodate the phase transition under the constraint of a constant decompression rate.